# Pathogenicity of *Avibacterium paragallinarum* Strains from Peru and the Selection of Candidate Strains for an Inactivated Vaccine

**DOI:** 10.3390/vaccines10071043

**Published:** 2022-06-29

**Authors:** Melanie Caballero-Garcia, Alfredo Mendoza-Espinoza, Silvia Ascanio, Paula Chero, Rober Rojas, Yosef Daniel Huberman

**Affiliations:** 1Research and Development, Quimtia S.A, Centro Ind. Las Praderas de Lurín Mza. B. Lote 1. Lima 16, Lurín CP 15823, Peru; melanie.caballero@quimtia.com (M.C.-G.); alfredo.mendoza@quimtia.com (A.M.-E.); silvia.ascanio@quimtia.com (S.A.); paula.chero.ancajima@gmail.com (P.C.); reyjas57@gmail.com (R.R.); 2Bacteriology, INTA EEA Balcarce, RN 226 Km 73, Balcarce CP 07620, Buenos Aires, Argentina

**Keywords:** *Avibacterium paragallinarum*, infectious coryza, pathogenicity, poultry, vaccine

## Abstract

Worldwide, *Avibacterium paragallinarum* is the aetiological agent of infectious coryza in poultry. Vaccines are the best means of control, helping reduce clinical signs and colonization of this bacterium. Most vaccines are based on international reference strains, or, lately, regional strains, but, generally, without any information regarding their virulence. The characterization of the pathogenicity of 24 *Av. paragallinarum* strains of the three Page serogroups, including four variant strains of serogroup B, all isolated from infectious coryza outbreaks in Peru, was performed. After experimental inoculation into the infraorbital sinuses, information regarding their capacity to induce infectious coryza typical clinical signs, spreading, and colonization was recorded. Furthermore, after intraperitoneal inoculation, septicaemia and death were registered. Differences among strains in these parameters were observed, even within strains from the same serogroup. Finally, the four most pathogenic strains, one from each serogroup, were chosen to formulate an experimental vaccine that was tested successfully against homologous challenges in reducing clinical signs and colonization in vaccinated birds compared to unvaccinated ones. This is the first time that *Av. paragallinarum* strains from Peru were studied thoroughly for their virulence in a search for improving vaccine formulation.

## 1. Introduction

Infectious coryza, caused by *Avibacterium paragallinarum*, is an acute respiratory disease that affects poultry worldwide [1]. Page classified this bacterium into three serogroups A, B, and C, according to the plate agglutination test [2]. Later, these schemes were amplified to recognize nine Kume serovars: A-1, A-2, A-3, A-4, B-1, C-1, C-2, C-3, and C-4 using hemagglutination inhibition tests [3,4]. Nevertheless, these schemes might need to be revised and updated as many B strains have been described as different [5,6,7,8]. Infectious coryza is characterized by inflammation of the upper respiratory tract as well as facial oedema, conjunctivitis, nasal discharge, diarrhoea, and anorexia, while wattles might also be affected [1]. There is no public health significance to *Av. paragallinarum,* and it is mainly reported in layers with a reduction of up to 40% in egg production, but it can get complicated when other pathogens are associated, increasing these losses up to 85% with considerable mortality rates [1,8,9,10]. Usually, clinical signs and reduction of egg production will disappear within 20 days while complicated cases may last up to two months [11]. On the other hand, infectious coryza has also been reported in broilers [8,12,13,14,15] usually associated with other bacterial or viral agents [9]. When complicated with other agents, arthritis and septicaemia have also been reported [8,12,16,17] and there is a specific description of how the coinfection with Fowl Adenovirus increased mortality in an experimental inoculation with serogroup A field strain of *Av. paragallinarum* [18].

The distribution of the three Page serogroups differs from country to country. Nevertheless, not all the countries report the distribution of the different serogroups. The three Page serogroups were reported in Argentina [13], Brazil [19], China [10,20,21], Ecuador [22], Egypt [23], Germany [24], India [25], Indonesia [26], Peru [5], Philippines [27], Mexico [28], South Africa [24,29], Spain [30], and the USA [2]. Furthermore, in Australia [31,32], Israel [33,34], and Japan [24,35], only serogroups A and C have been reported, while in Zimbabwe, only serogroups B and C [7,36] are known. In Malaysia [37], only serogroup A, in Panama and Thailand only serogroup B [34,38], while in Taiwan only serogroup C [39] were described. Moreover, some serogroup B strains were reported in Argentina, Ecuador, Peru, the USA, and Zimbabwe as variants (Bvar), and, as above-mentioned, might represent a new immunotype [5,7,9]. The diagnosis of classic infectious coryza is performed by bacteriological culturing of mucus samples taken from the infraorbital sinus or the nasal turbinates. In septicaemia cases, lungs, air sacs, livers, and spleens can also be examined. *Av. paragallinarum* has been isolated even from inside eyeballs in acute cases. Columbia agar base plates plus 7% equine hemolyzed blood [13] may be used with or without the addition of antibiotics, such as bacitracin, cloxacillin, or vancomycin, to eliminate containment bacteria [40]. The molecular diagnosis of mucus obtained from squeezing the sinus of live birds was equivalent to culture, but much more rapid [41].

To protect the chickens against *Av. paragallinarum* infections, spreading, and reduce economic losses, good management practices and strict biosecurity measures should be applied, such as in the case of other diseases. The use of some antibiotics may reduce clinical signs, but the birds remain carriers for life, and, even more, *Av. paragallinarum* is likely to develop resistance [1,42,43]. Alternative to antibiotics, a significant reduction of clinical signs was reported by using disinfectants in drinking water and spraying only in vaccinated birds [44,45] but this will not eliminate the presence of *Av. paragallinarum*. Hence, a good vaccination program is very efficient, as inactivated vaccines are very protective when they are well prepared [1]. Generally, homologous protection within Page serogroups is maintained [46,47,48], although it might be reduced within the B strains [49,50]. Furthermore, within Kume serogroups, varying levels of cross-protection were observed using nine reference strains [51].

The pathogenicity of *Av. paragallinarum* strains is variable and was evidenced using field and reference strains [52,53]. These differences might lead to reduced protection given by bacterins that do not include local or regional pathogenic strains. Different strains of serogroup B were reported to provide only partial cross-protection among themselves, suggesting the need to include more than one B serogroup [7,49]. Similar findings of incomplete cross-protection were also reported within the Kume C serogroup [51,54]. Nevertheless, there is no information correlating the pathogenicity of a certain strain with its capacity to protect against homologous or heterologous challenges.

Therefore, it is suggested that the inclusion of *Av. paragallinarum* strains in inactivated vaccines might be improved after being selected not only for their serotype but also according to their pathogenicity. For that purpose, the classification of local Peruvian stains of *Av. paragallinarum* was performed to study their capacity to cause clinical signs, horizontal infection as well as cause septicaemia and death in experimentally inoculated chickens. Afterwards, the most pathogenic strains of serogroups A, B, Bvar, and C were included in an experimental bacterin, and its efficacy was tested *in vivo* against homologous experimental infection with the same four serogroups.

## 2. Material and Methods

### 2.1. Bacterial Strains

A total of 24 strains of *Av. paragallinarum*, isolated from poultry farms in Peru, were used (Appendix A). These strains were isolated from chickens that suffered infectious coryza outbreaks and kept frozen at −80 °C at the bacterial collection of QUIMITA SA (Lima, Peru) until usage. All strains were tested by biochemical assays typical for this species: motility, TTC, nitrate reduction using the Griess–Islovay reagent, Urease, indol, and carbohydrate fermentation tests were performed as previously described and, finally, confirmed by PCR [41]. All strains were classified as reduced β-Nicotinamide Adenine Dinucleotide (NAD) dependent, except for strain Q23.

Furthermore, these strains were serotyped and classified into the three Page serogroups using inhibition of the haemagglutination assays [4] using rabbit antiserum raised against international reference strains 0083, 0222, and Modesto strains of *Av. paragallinarum* strains of serogroup A, B, and C, respectively. Additionally, strains Q14–Q17 were serotyped using antiserum raised against Argentinean *Av. paragallinarum* strains H23 (serogroup A), H8 (serogroup B), and H32 (serogroup C), as it was not possible to determine their serogroup using the international reference antiserums and were regarded as variants of serogroup B (Bvar) [5,44,50]. Strain Q7 showed similar inhibition of the haemagglutination using antisera of serogroups A and B, both of the international reference strains as well as the Argentinean strains, and was classified as serogroup A/B. For the selection of strain in the present work, this strain was arbitrarily considered serogroup A.

### 2.2. Avianization and Preparation of Inoculum

Before each trial, the corresponding avianized *Av. paragallinarum* strain was thawed from the frozen vial and was streaked onto Columbia Blood agar base (Oxoid, CM0331, Basingstoke, Hants, UK) + 7% equine hemolyzed blood (CLBA). These plates were incubated for 48 h at 37 °C in a microaerophilic atmosphere using the candle jar method. Afterwards, a few isolated colonies were used to seed a Brain Heart Infusion (BHI) broth (Merck 1.10493, Darmstadt, Germany) supplemented with 25 µg/mL of reduced NAD (Sigma, N8129, St. Louis, MO, USA). This tube was incubated as above and before inoculation, bacterial enumeration was performed using triplicate CLBA plates [55].

To increase the virulence of the strains [11] and standardize the proceedings, each one of the 24 strains was passaged *in vivo* by inoculation of 0.2 mL of the correspondent inoculum into the infraorbital sinus of two 25-days-old SPF chickens. Two days after the inoculation, upon the manifestation of clinical signs, the birds were euthanized, each strain was re-isolated, and the growth was aliquoted and kept frozen (−80 °C) until usage.

### 2.3. Chickens

One-day-old Hy-Line layers chicks were used (Produss, Lima, Perú). *Salmonella*-free status was confirmed by bacteriological analysis of the meconium of all chicks. These samples were pooled together into 100 mL Tetrathionate Broth (TB) (Merck 1.05285), followed by subcultures onto XLD (Difco 278850, Sparks, MD, USA) + 0.46% Tergitol 4 (Sigma 100H0494) (XLDT4) agar plates. *Salmonella*-like colonies were further checked by biochemical tests to discard the presence of *Salmonella* sp.

### 2.4. Housing, Feed, and Water

All chicks were reared until the beginning of each trial under strict biosecurity measures. During the first week of life, all birds were individually identified with numbered wing tags. The distribution of the chickens into the experimental groups, as well as the selection of the isolator, were random and coded in a manner that the operators or investigators were blind to treatments. Each isolator was used to house no more than 10 chickens. The rearing facility consisted of a closed room with a ventilation system and automatically controlled temperature and lighting systems. Before entering the building, all personnel had to take a shower and used disposable gloves, hats, masks, clean clothes, and boots. Each isolator has glassed front doors and a wire net floor to avoid contact with faeces. Water was administered by nipples and feed was given *ad libitum* from the outside through plastic pipes into the feeders. Animals received *Salmonella*-free balanced food based on a vegetable protein diet, free from meat or fishmeal, without adding any antibiotic or coccidiostat drug. The absence of *Salmonella* spp. in feed was confirmed by bacteriological cultures of all batches using Lactose broth followed by a subculture into TB and XLDT4 agar plates as previously described [56].

## 3. Experimental Design

Four sets of trials were performed: the first two sets of trials were done to investigate the pathogenicity of all strains regarding their capacity to cause clinical signs after inoculation (Trial 1) and their capacity to horizontally infect non-inoculated chickens (Trial 2). The capacity to colonize the inoculated infraorbital sinuses was also studied in both trials by bacteriology cultures.

The third set of trials was performed with the most pathogenic strains that were selected from Trials 1 and 2 to learn the capability to cause septicaemia after intraperitoneal inoculation and its lethal capacity. Finally, after performing a qualitative analysis of the results from the three trials, the most pathogenic strains, one for each serogroup, were selected for the elaboration of an experimental bacterin that was tested for efficacy against homologous challenges with the same strains. A description of the experimental design is available in Appendix A.

In all trials, similar procedures were used. The chickens were distributed into two isolators for each experimental group. The inoculations were done by injection of 0.2 mL of the correspondent inoculum into the left infraorbital sinus of each chicken. Clinical-signs scores of 0–4 were registered individually for each chicken. In Trials 1 and 4, on day 5 post-inoculation, all chickens were euthanized, heads were separated from the body and a sterile cotton swab, previously moistened in sterile BHI, was introduced separately into each of both infraorbital sinuses and was immediately streaked onto CLBA plates that were incubated as before. Differently, in Trial 2, inoculated chickens were housed with non-inoculated birds in the same isolator. In this set of trials, the registration of clinical signs scores to all birds was performed on days 2, 5, and 7 post-inoculation. Euthanasia of birds was done on day 7 post-inoculation, followed by bacteriological culturing as above. On the other hand, Trial 3 was performed by intraperitoneal infections, and mortality was registered until day 3 post-inoculation before all birds were euthanized. Livers from dead and euthanized birds were directly cultured on CLBA plates that were followed as before.

The number of strains that were tested at the same time was determined according to the total number of isolators in the experimental unit. During all four trials, in each experiment, two experimental groups of 10 chickens each were added; in the first group, the chicks were inoculated with sterile BHI, while in the other, 10 birds remained as the negative control. These 20 birds were housed at the same facilities under the same conditions and had to be negative for clinical signs and stay free from *Av. paragallinarum* infections at all times to maintain the validity of the results.

Trial 1 was performed using 20 chickens of 20 days of age for each experimental group, housing 10 birds in each isolator. Birds were inoculated by depositing 0.2 mL of the correspondent inoculum into the infraorbital sinus. The clinical signs score was registered on days 2 and 5 post-infection and bacteriological examinations were performed on euthanized birds on that day.

Trial 2 was performed by inoculating four chicks, as in Trial 1, and housing them together with another six chicks in close contact within the same isolator. Clinical signs score was registered on days 2, 5, and 7 post-infection and bacteriological examination was performed on euthanized birds on that day.

According to the pathogenicity and horizontal infection results from Trials 1 and 2, twelve strains were chosen: three from serogroup A, two from serogroup B, four from serogroup Bvar, and three from serogroup C. These strains were used in Trial 3 where 20 seven-days-old birds for each strain were inoculated intraperitoneally with 0.5 mL of the correspondent inoculum and mortality was observed during 72 h post-infection. Bacteriological examination of the livers of the dead birds as well as of euthanized surviving chickens was performed.

### 3.1. Preparation of the Experimental Vaccine

The most pathogenic strains from Trials 1, 2, and 3 were selected. For each strain, a few colonies from CLBA plates of these strains were used to seed BHI supplemented with reduced NAD, which was incubated for 48 h at 37 °C. Afterwards, thimerosal was added (0.01%) and left at 2–8 °C for inactivation for 14 days. Inactivation was confirmed by bacteriological culturing. The final bacterial concentration was 5 × 10^8^ for each strain. Calcium Phosphate was added as an adjuvant at a final concentration of 0.7%.

### 3.2. Protection Trial

Vaccination of four experimental groups was done at weeks 5 and 8 of life. The vaccine was administered by subcutaneous injection of 0.5 mL of the vaccine at the back of the neck. Each experimental group consisted of 12 chickens and a similar number of unvaccinated chickens was used as the negative control for each of the four selected most pathogenic strains. As above-mentioned, another group of 10 birds was added as non-vaccinated and non-inoculated chickens for internal control. The infection trials were performed three weeks after the second vaccination dose comparing clinical signs scores and bacteriological re-isolation rates between vaccinated and unvaccinated chickens. Clinical signs scores were registered on days 2 and 5 post-infection and bacteriological examinations were performed on euthanized birds on that day. In this trial, a bird was considered diseased if it scored at least 2 points on one of both days of observations of clinical signs or if *Av. paragallinarum* was re-isolated from one or two infra-orbital sinuses.

### 3.3. Evaluation of Clinical Signs

In all trials, clinical signs of each facial side were registered according to the following criteria: 0, no clinical signs; 1, light conjunctivitis; 2, facial inflammation, infraorbital sinusitis, and/or conjunctivitis and/or tears; 3, oedema and facial inflammation, infraorbital sinusitis, abundant nasal and eye secretion, and partially closed eye; and 4, severe oedema and facial inflammation, abundant nasal and eye secretion, and complete closure of the eye. Scores of 0 and 1 were considered negative, while scores of 2, 3, or 4 were considered positive [40,50].

### 3.4. Interpretation of Results

In trials 1 and 2, each bird was given a score between 0 and 4 points for each infraorbital sinus, with a theoretical maximum of 8 points per bird on each observation day. Additionally, bacteriological examination for each sinus for each infraorbital sinus was also considered as an additional theoretical maximum of 2 points. This way, in Trial 1, each bird could sum up to 18 points while in Trial 2, each bird could sum up to 26 points (as there was one more day of observations for clinical signs). The total number of points from Trials 1 and 2 was used to select the more pathogenic strains of each serogroup to be further studied. This way, 12 strains were chosen for Trial 3, where a dead bird was given 1 point for each strain and the re-isolation of *Av. paragallinarum* from the liver gave an additional point. The total number of points from Trial 3 together with the previous ones was used to select the most pathogenic strains to formulate the experimental bacterin and as challenge strains for Trial 4.

### 3.5. Statistics

Each animal was considered as the experimental unit, with the only exception of the horizontal infection trial (Trial 2) where the six non-infected chickens were considered as the experimental unit.

In Trials 1, 2, and 3, the number of chickens was based on previous trials that described the pathogenicity of *Avibacterium paragallinarum* strains [25,57,58]. In these works, each experimental group consisted of 10 chickens. As no statistical differences were evidenced by these authors, it was decided to duplicate this number, using two groups of 10 chickens each. In Trial 4, the number of chickens was defined by using a minimum infective dose that causes 100% of infection in the control group expecting a reduction of 50% in the vaccinated groups (with 95% of confidence, potency power of 90%). Therefore, 12 chickens were divided into two isolators of six chickens each. This number is similar to other works that used 10 birds in challenge trials [7,52].

Nevertheless, the results of Trials 1 and 2 were evaluated without any statistical interpretation based on qualitative information about the presentation of clinical signs (categorical results). Within the experimental groups in these two trials, there were no statistical differences regarding the re-isolation rates of *Av. paragallinarum*.

In Trial 3, the chi-square test of independence was performed for the results obtained comparing the number of dead birds or the number of positive livers within strains of the same serogroup with a level of significance of 95%.

In Trial 4, the Fisher Exact test was used to compare the number of birds with clinical signs, the number of positive infraorbital sinuses, and the number of diseased birds comparing vaccinated birds in comparison with non-vaccinated ones, for each serogroup with a level of significance of 95%.

### 3.6. Animal Welfare

Handling of birds was performed according to the Committee for the Update of the Guide for the Care and Use of Laboratory Animals [59] and euthanasia was performed by decapitation according to the American Veterinary Medical Association’s Manual for euthanasia [60]. The performance of these trials was previously evaluated and approved by the local Ethical Committee for the Use of Animals in Investigations (approval #2018-LIA020).

## 4. Results and Discussion

Worldwide, infectious coryza affects poultry and it is a cause of important economic losses in layers as well as in broilers. Nevertheless, the employment of good biosecurity measures together with good vaccination programs should be sufficient to protect against the disease when the bacterin is adequately prepared [1]. Most of the first commercial vaccines against the disease were prepared with only one single strain that was specific to the needs of localized/regional poultry operations. Nevertheless, these vaccines did not protect against other serogroups [61]. Later, some vaccines were prepared with serogroups A and C, assuming erroneously that serogroup B was not pathogenic and there was no need for its inclusion or that these two serogroups were efficient enough to cross-protect against serogroup B infections [49,50,62]. Even in trivalent vaccines, this low, or no protection against B strains was thought to be due to antigenic differences between local B strains and the international reference strains that were included in the vaccines. Therefore, it was concluded that B strains should be included onwards in vaccines.

A good vaccine should be composed of local/regional strains and should include at least one strain from each serogroup [1]. Additionally, the inclusion of variant B strains in the vaccines in addition to international known strains [7] was shown to be efficient and the incorporation of more than one strain of serogroup C helped to also cross-protect the chickens [63]. In the present study, taking into account the presence of Bvar strains in Peru [5] in addition to Page serogroups A, B, and C, the experimental vaccine was also composed of Bvar serogroup. Furthermore, a correct selection of *Av. paragallinarum* strains to be included in the vaccines should contain some facts regarding their pathogenicity and capability to cause clinical signs, spread to non-infected birds, and cause septicaemia and death of experimentally inoculated birds. This information may be added to the ability of the strains to colonize the inoculated infraorbital sinus as well as the not-inoculated one.

All hens remained healthy and free from any external infection until challenged. In every inoculation trial, two experimental groups of 10 chickens each were added; in the first group, the chicks were inoculated with sterile BHI, while in the other, 10 birds remained as the negative control. The trial was considered valid only if none of these birds demonstrated clinical signs and stayed free from infection by *Av. paragallinarum*.

In the present work, the pathogenicity of 24 Peruvian *Av. paragallinarum* strains was studied. These strains were collected from clinical cases in Peru during the last 20 years. As the pathogenicity of *Av. paragallinarum* might be altered upon long conservation and an undefined number of passages in artificial culture media, all strains were passaged in live birds, re-isolated, and kept with a standardized number of subcultures [11,14]. This way, all inoculation trials were performed under the same conditions.

Later, inoculation studies were performed comparing their capacity to cause clinical signs (Trials 1 and 2) and to spread to non-infected birds (Trial 2). The use of clinical signs score for pathogenicity studies was used by other authors, although criteria may vary, it is a good way to describe the course of infection and its severity [14,44,57,58,64,65,66,67]. The results of clinical signs scores, bacteriology, and deaths from Trials 1, 2, and 3 are available in Table 1 and in Appendix A. In general, as expected, clinical signs scores were notably higher on day 2 post-inoculation in comparison with days 5 or 7, while low scores were registered for the non-inoculated facial sides. Some strains were responsible for 100% morbidity while in others, low morbidity was registered with a variation in the extent of the clinical signs among the chickens. In Trial 1, for example, within serogroup A strains, some strains had scored 49 or 50 points per strain (Q7 and Q2, respectively), while others only 2 or 3 per strain (Q6 and Q4, respectively), practically not presenting clinical signs. Although not that extreme, a gradient of clinical signs scores was also observed among strains of serogroup B (total scores between 33 and 69 per strain) and C (total scores between 31 and 96 per strain). Nevertheless, almost no differences were registered in these scores among the four strains from the Bvar strains (61–68). On the other hand, for all serogroups, bacteriological results were more even; *Av. paragallinarum* was re-isolated from all inoculated infraorbital sinuses and almost from all the non-inoculated ones and varied between 36 and 40.

Strains Q10 and Q13 from serogroup B were not lethal as well as strain Q24 of serogroup C, with only 2, 3, and 4 positive livers, respectively. Mortality rates varied among the other nine strains: strains Q2 and Q17 resulted in 7 and 6 dead birds, respectively, and were the two strains with more dead birds than all the others. Within serogroup Bvar, these 6 dead birds (Q17) were significantly more than the only 1 dead bird for each of the strains Q15 and Q16 (*p* < 0.05).

In Trial 2 (Appendix A), besides the registration of clinical signs for inoculated birds, clinical signs were also observed in non-infected ones that were reared in close contact, in the same isolator, with the inoculated birds. Observations on days 2, 5, and 7 post-inoculation resulted, as in Trial 1, in a gradient of the total clinical signs scores among the strains. Furthermore, these totals varied within the same serogroups: for serogroup A, from 39 to 87; for serogroup B, from 17 to 59; and for serogroup C, from 11 to 64; while for serogroup Bvar there was a relatively lower variation, from 42 to 75. A greater variation was observed regarding bacteriological re-isolations of *Av. paragallinarum* with some surprise to isolate the bacterium from groups with almost no clinical signs, such as strain Q19, demonstrating colonization capacity but low pathogenicity. Furthermore, some strains with high bacteriological results from Trial 1 had a much lower number of positive samples in Trial 2, such as strains Q8 and Q12 with only 14 and 8 positive samples, respectively. These strains were not as persistent and were eliminated by the birds when sampled on day 7 post-inoculation. Apparent healthy birds might likely pass unnoticed, but carrier birds have long been recognized as the main reservoir of Infectious coryza infection [1].

Outbreaks with high mortality and septicaemia caused by *Av. paragallinarum* have also been reported and it was isolated from the liver and kidney as well in other sites [8]. *Av. paragallinarum* was also reported to be involved with peritonitis [64]. In Argentina, intraperitoneal experimental infection with pathogenic strains of the three serogroups has demonstrated the existence of non-lethal stains, while others caused more than 90% mortality [62]. Therefore, the capacity to cause septicaemia and death by the strains in the present study was investigated as well. For that purpose, the more pathogenic strains were selected according to the total clinical signs scores and bacteriological results from Trials 1 and 2. This way, 12 strains were chosen and birds were inoculated intraperitoneally. For serogroup A, strains Q2, Q3, and Q7 were selected as well as strains Q10 and Q13 for serogroup B, the four strains of serogroup Bvar Q14–Q17, and strains Q18, Q21, and Q24 for serogroup C.

On the other side, the re-isolation of the challenge strains was significantly higher in strains Q2, Q7 (both from serogroup A), Q12, and Q18 (serogroup C) with 18, 18, 18, and 12 positive livers, respectively, in comparison with the other strains of the same serogroup. Within serogroup Bvar, strains Q17, Q16, and Q14 had, respectively, 11, 7, and 6 positive livers, significantly more (*p* < 0.05) than strain Q15 with only one.

Summing up these results, four strains, one for each serogroup, were selected to formulate the experimental inactivated vaccine: strains Q2, Q13, Q17, and Q18, respectively for serogroups A, B, Bvar, and C. This vaccine was tested to evaluate protection against homologous challenges with the same serogroups using the vaccination-challenge model, which is regarded as the “gold standard” to determine the immunological response in chickens that have been vaccinated with a double dose of *Av. paragallinarum* [68]. Chickens were vaccinated twice and challenged separately with each one of these four strains. No adverse effects, such as a change of behaviour or local reaction at the vaccination site, were observed in the vaccinated chickens. Good protection was registered for the four strains that were used. Clinical signs were notably reduced and significantly lower re-isolation rates of *Av. paragallinarum* and the number of positive chickens were found in vaccinated birds in comparison with the unvaccinated ones (Table 2).

## 5. Conclusions

The present work is the first that studied the pathogenicity of *Av. paragallinarum* strains isolated from poultry in Peru. A set of pathogenicity trials of these strains, regarding their capacity to cause clinical signs, horizontally infect susceptible birds, and their capacity to cause septicaemia and death was performed. A great variation among these parameters was observed, even within strains of the same serogroup. Afterwards, the most pathogenic strains for each serogroup were selected to formulate an experimental multivalent inactivated vaccine and a vaccination-challenge trial was performed. Homologous protection with satisfying levels of protection was observed, reducing clinical signs and significantly reducing the colonization of the infraorbital sinuses after infection, reducing the number of carrier birds.

The results of the present trials suggest an alternative methodology for the selection of *Av. paragallinarum* strains to be included in a vaccine according to their pathogenicity and not only by their serotype. Future trials should include a performance comparison of the experimental vaccine with commercial vaccines (positive control) and test the efficacy in a complete vaccination program in layers during production under controlled conditions or in field trials [69]. If field trials are successful, this methodology might, and should, be adopted in other regions to improve vaccine performance on a local/regional basis, especially where infectious coryza is prevalent.

## Figures and Tables

**Table 1 vaccines-10-01043-t001:** Trials 1, 2, and 3. Summary of clinical signs scores, re-isolation rates of *Avibacterium paragallinarum* from infraorbital sinuses (Trials 1 and 2) or livers (Trial 3), and death caused after intraperitoneal inoculations (Trial 3). In Trial 3, only the most pathogenic strains from Trials 1 and 2 were tested. More detailed results are available in Appendix A.

Serogroup	Strain	Trial 1. Pathogenicity			Trial 2. Horizontal Infection			Trial 3. Septicaemia and Death		
		CS	Bact.	ST	CS	Bact.	ST	Death	Bact.	ST
A	Q1	20	38	58	59	38	97	-	-	-
	Q2	50	38	88	87	28	115	7	18	25
	Q3	36	38	74	75	32	107	2	8	10
	Q4	3	40	43	74	36	110	-	-	-
	Q5	10	39	49	53	33	86	0	0	0
	Q6	2	40	42	39	28	67	-	-	-
	Q7	49	40	89	68	31	99	4	18	22
B	Q8	53	40	93	17	14	31	-	-	-
	Q9	33	37	70	36	26	62	-	-	-
	Q10	59	37	96	45	21	66	0	2	2
	Q11	41	38	79	41	23	64	-	-	-
	Q12	43	40	83	19	8	27	-	-	-
	Q13	69	40	109	59	34	93	0	3	3
Bvar	Q14	61	39	100	42	37	79	3	6	9
	Q15	67	40	107	66	35	101	1	1	2
	Q16	61	39	100	75	38	113	1	7	8
	Q17	68	40	108	57	32	89	6	11	17
C	Q18	83	37	120	29	36	65	4	18	22
	Q19	36	40	76	18	35	53	0	5	5
	Q20	58	39	97	13	17	30	-	-	-
	Q21	96	36	132	21	35	56	5	12	17
	Q22	60	40	100	11	20	31	-	-	-
	Q23	31	39	70	64	37	101	-	-	-
	Q24	59	39	98	23	18	41	0	4	4

CS = Clinical Signs; Bact. = Re-isolation rate of *Av. paragallinarum*; and ST = Subtotal.

**Table 2 vaccines-10-01043-t002:** Trial 4. Protection. Chickens were vaccinated subcutaneously on weeks 5 and 8 of life. Three weeks later, all birds were inoculated into the left infra-orbital sinus with 0.2 mL of each strain. Clinical signs score for all birds were observed on days 2 and 5 post-inoculation. The birds were scored individually between 0 and 4 points for each facial side [58]. A bird was considered positive if it scored at least 2 points in one infra-orbital sinus. On day 5 post-inoculation, all birds were euthanized and both infra-orbital sinuses were cultured for the presence of *Avibacterium paragallinarum*. A bird was considered diseased if it scored at least 2 points on one of both days of observations for clinical signs or if *Av. paragallinarum* was re-isolated from one or two infra-orbital sinuses.

Strain (Serogroup)	Dose (UFC/mL)	Vaccine	n	Total Clinical Signs Scores (Days Post-Inoculation)		Number of Positive Birds (Days Post-Inoculation)		Presence of *Av. paragallinarum*		Number of Diseased Birds
				2	5	2	5	Left	Right	
Q2 (A)	1.95 × 10^7^	Yes	12	10	4	3 *	0*	1 *	0 *	3 *
		No	12	42	36	12	11	12	8	12
Q13 (B)	1.9 × 10^7^	Yes	12	6	3	0 *	0*	0 *	0*	0 *
		No	12	31	23	12	6	12	8	12
Q17 (Bvar)	6.2 × 10^7^	Yes	12	9	2	1 *	0*	0 *	0 *	1 *
		No	12	34	24	12	9	12	10	12
Q18 (C)	6 × 10^7^	Yes	12	12	5	2 *	0*	2 *	0 *	2 *
		No	12	35	23	12	7	12	9	12

* Statistically lower than the non-vaccinated control group that was challenged with the same strain. Fisher exact test *p* < 0.05.

## Data Availability

Not applicable.

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
