# Peer review of "Pathogenicity of Avibacterium paragallinarum Strains from Peru and the Selection of Candidate Strains for an Inactivated Vaccine"

_vaccines, 2022, doi:10.3390/vaccines10071043_

Round 1

Reviewer 1 Report

Thank you for sending me the research article paper “Pathogenicity of Avibacterium paragallinarum Strains from 2 Peru and the Selection of Candidate Strains for an Inactivated 3 Vaccine” for review in the Vaccines. In the article of Melanie et al., the author discussed the pathogenicity of Avibacterium paragallinarum Strains and its vaccinated capability. There are important points that should be discussed and improved.

1.                   Author should mention the current status of Avibacterium paragallinarum treatment and diagnosis.

2.                   Author should present the result related to the confirmation of bacterial strain e.g., PCR and biochemical assay. Furthermore, another confirmation assay related with the isolation of bacteria from Columbia blood agar.

3.                   What is the detailed protocol of vaccine preparation and its confirmation?

4.                   It would be good to present the experimental design in the form of flow diagrammed.

5.                   It would be great to provide the behavioral analysis of animals on vaccine inoculation.

Author Response

Reviewer 1

Thank you for sending me the research article paper “Pathogenicity of Avibacterium paragallinarum Strains from Peru and the Selection of Candidate Strains for an Inactivated Vaccine” for review in the Vaccines. In the article of Melanie et al., the author discussed the pathogenicity of Avibacterium paragallinarum Strains and its vaccinated capability. There are important points that should be discussed and improved.

  1. Author should mention the current status of Avibacterium paragallinarum treatment and diagnosis.

Treatments and the use of antibiotics and disinfectants are described (lines 76-79) of the revised manuscript with references to previous works and information about the diagnosis was added as requested (lines 62-71 of the revised manuscript)

  1. Author should present the result related to the confirmation of bacterial strain e.g., PCR and biochemical assay. Furthermore, another confirmation assay related with the isolation of bacteria from Columbia blood agar.

Information about PCR and Biochemical assays was added as requested (lines 105-113 of the revised manuscript).

  1. What is the detailed protocol of vaccine preparation and its confirmation?

Detailed information about vaccine preparation is given under the title “Preparation of the experimental vaccine” (lines 224-230 of the revised manuscript)

  1. It would be good to present the experimental design in the form of flow diagrammed.

The authors accept this suggestion and added Figure S1.

  1. It would be great to provide the behavioral analysis of animals on vaccine inoculation.

No behavioural data was registered during the trials. The text was modified to clarify: “No adverse effects like the change of behaviour or local reaction at the vaccination site, were observed in the vaccinated chickens.”  (lines 412 - 414 of the revised manuscript).

Reviewer 2 Report

The introduction is really longer than necessary. Whilst the information is useful, I suggest that some basic descriptions (eg clinical signs of the infection) would be deleted, as they can be found in standard textbooks and are well known by potential readers of this manuscript.

The hypothesis of the authors must be clearly described in a new paragraph, immediately before the end of the section.

Table 1. Please move to supplementary material. You can add a map with the locations of the farms, where the isolates were obtained.

Please described in greater detail the inoculation technique of the chicks.

Please include a new table summarizing the experimental design, which will be very useful to fully understand the study.

Same for the protection trial

Also, a new table for the scores and the interpretation.

Table 2. Too detailed, please move to supplementary material.

Same for tables 3 to 6.

Please include graphs, with summaries of results.

In general, I do not like to merge results with discussion, but I can understand the authors.

However, there is a need to add an overall section for summarizing the discussion at the end.

The manuscript can advance to the next stage. However, there is need for extensive changes to be made with great attention, before resubmission.

The manuscript needs further evaluation after revision.

Author Response

Reviewer 2

  1. The introduction is really longer than necessary. Whilst the information is useful, I suggest that some basic descriptions (eg clinical signs of the infection) would be deleted, as they can be found in standard textbooks and are well known by potential readers of this manuscript.

Some text from the Introduction was deleted as suggested. Nevertheless, despite the references that are mentioned in the text for treatments and diagnosis, according to comments that were made by other reviewers of this manuscript, more information was added in a manner that the introduction is not shorter than before.

  1. The hypothesis of the authors must be clearly described in a new paragraph, immediately before the end of the section.

Added as suggested: “Therefore, it is suggested that the inclusion of Av. paragallinarum strains in an inactivated vaccine might be improved after being selected not only for their serotype but also according to their pathogenicity”. (lines 94-96 of the revised manuscript).

  1. Table 1. Please move to supplementary material. You can add a map with the locations of the farms, where the isolates were obtained.

Modified as suggested. The table now appears as supplementary material in Table S1.

  1. Please described in greater detail the inoculation technique of the chicks.

More information was added as requested: “0.2 mL of the correspondent inoculum” (lines 138-139 of the revised manuscript).

  1. Please include a new table summarizing the experimental design, which will be very useful to fully understand the study.

As was suggested by another reviewer a diagram was added as supplementary material in Figure 1S.

  1. Same for the protection trial

The table was modified to improve understanding as requested by other reviewers.

  1. Also, a new table for the scores and the interpretation.

Instead of a new table, a paragraph was added, “Evaluation of clinical signs” for that purpose (lines 246-253 of the revised manuscript).

  1. Table 2. Too detailed, please move to supplementary material. Same for tables 3 to 6. Please include graphs, with summaries of results.

These tables were moved as suggested and detailed information is presented as Supplementary material (Tables S2, S3, S4, S5).

Reviewer 3 Report

The manuscript describes the development and evaluation of inactivated vaccines to Avibacterium paragallinarum. This review was restricted to the Material and Methods (M&M) section, whereas the full manuscript will be subsequently reviewed upon a response to my concerns related to the the experimental design. Specifically, the experimental design needs to be re-written for clarity. In that context, the re-written text can also be supplemented with a table(s) to further illustrate the experimental design. However, the primary objection concerns the apparent absence of duplicate experimental groups among the experimental groups in all trials. Of course, each experimental exposure for each experimental group should be at least performed in duplicate; this is standard experimental procedure, but is not clear in the narrative of the experimental design in the M&M. If the experiments were not performed in duplicate for each experimental group in all trials, then I can only reject the manuscript for publication. If this is incorrect, please explain in the text/tables with clarity and resubmit for review.

Author Response

Reviewer 3

The manuscript describes the development and evaluation of inactivated vaccines to Avibacterium paragallinarum. This review was restricted to the Material and Methods (M&M) section, whereas the full manuscript will be subsequently reviewed upon a response to my concerns related to the the experimental design. Specifically, the experimental design needs to be re-written for clarity. In that context, the re-written text can also be supplemented with a table(s) to further illustrate the experimental design. However, the primary objection concerns the apparent absence of duplicate experimental groups among the experimental groups in all trials. Of course, each experimental exposure for each experimental group should be at least performed in duplicate; this is standard experimental procedure, but is not clear in the narrative of the experimental design in the M&M. If the experiments were not performed in duplicate for each experimental group in all trials, then I can only reject the manuscript for publication. If this is incorrect, please explain in the text/tables with clarity and resubmit for review.

Information regarding the distribution of the birds in experimental groups is described in original manuscript. The authors wish to make it clear that each experimental group was divided into 2 isolators. Nevertheless, as each animal was considered as the experimental unit (including the only exception of the horizontal infection in Trial 2 where the six non-infected chickens were considered as the experimental unit), there were enough repetitions for the different treatments. The results of Trials 1 and 2 were evaluated without any statistical analysis based on qualitative information about the presentation of clinical signs

To ensure easier and clear interpretation, more information was added:

Housing, Feed, and Water – “Each isolator was used to house no more than 10 chickens.(Lines 155-156 of the revised manuscript).

Experimental design – “The chickens were distributed into two isolators for each experimental group.” (Lines 182-183 of the revised manuscript).

“Finally, after performing a qualitative analysis of the results from the three trials, the most pathogenic strains, one for each serogroup, were selected for the elaboration of an experimental bacterin that was tested for efficacy against homologous challenges with the same strains.(Lines 176-180 of the revised manuscript).

Statistics – “Each animal was considered as the experimental unit with the only exception of the horizontal infection trial (Trial 2) where the six non-infected chickens were considered as the experimental unit. Nevertheless, the results of Trials 1 and 2 were evaluated without any statistical interpretation based on qualitative information about the presentation of clinical signs (categorical results). Within the experimental groups in these two trials. There were no statistical differences regarding the re-isolation rates of Av. paragallinarum.” (Lines 269-275 of the revised manuscript).

Reviewer 4 Report

General comments:

The manuscript describes a series of trials testing the pathogenicity of Avibacterium paragallinarum strains isolated from infected chicken in Peru, as well as a vaccine efficacy trial testing the protection conferred by a vaccine formulated with the 4 most pathogenic strains identified. The trials have been well designed and the manuscript well written.

My main critique of the paper is that the vaccine efficacy trial does not include a group vaccinated with the already in use commercial vaccine as a control group. Although a non-inferiority trial may have been out of scope for this paper it would have been interesting to see. This should at least be discussed as a possible next step.

Some discussion on the choice of using homologous challenges only, instead of heterologous, would also be interesting to read.

And do you think this new vaccine containing multiple highly virulent strains should be tested globally, or is it intended more for use in Peru alone?

Specific comments:

Lines 29-34: Introduce Page and Kume serogroups here, it confuses readers who are not familiar with Avibacterium paragallinarum when these terms are used later in the text.

Lines 160-162: Kindly add a description of the criteria used for the different scores

Lines 167-169; 183; 188, 214: The wording is confusing, how can clinical scoring be done on days 2, 5 and 7, and the birds also killed? If they were killed on day 2 they can’t be clinically scored on day 5 or 7…. I think the confusion lies in that the days of euthanasia have not been clarified. And then I think you mean that the bacteriology was performed directly on the day of euthanasia. Please add day of euthanasia (and humane-endpoint-criteria if used), and rewrite these sentences more clearly.

Line 190: translate Spanish “y” to English “and”. 

Line 201: change “an” to “and”

Line 202: remove “A”

Lines 214-215: please change: “a diseased bird was considered if” to “a bird was considered diseased if”

Line 245: correct “infection coryza”

Table 2:  please add number of birds per group, at least in the table legend, and number of dead animals per group

Table 6: please define “number of positive birds”.

Author Response

Reviewer 4

The manuscript describes a series of trials testing the pathogenicity of Avibacterium paragallinarum strains isolated from infected chicken in Peru, as well as a vaccine efficacy trial testing the protection conferred by a vaccine formulated with the 4 most pathogenic strains identified. The trials have been well designed and the manuscript well written.

  1. My main critique of the paper is that the vaccine efficacy trial does not include a group vaccinated with the already in use commercial vaccine as a control group. Although a non-inferiority trial may have been out of scope for this paper it would have been interesting to see. This should at least be discussed as a possible next step. Some discussion on the choice of using homologous challenges only, instead of heterologous, would also be interesting to read.

The authors agree with this comment. Indeed, in the near future, these trials are planned to be conducted as suggested including field trials in henhouses with a history of Infectious Coryza. A text was added as recommended:

“Future trials should include a performance comparison of the experimental vaccine with available commercial vaccines (positive control) and to test the efficacy and in a complete vaccination program in layers during production under controlled conditions or in field trials.” (lines 439-442 of the revised manuscript).

  1. And do you think this new vaccine containing multiple highly virulent strains should be tested globally, or is it intended more for use in Peru alone?

This is a very interesting point for discussion. The following text was added:

“These trials suggest an alternative methodology for the selection of Av. paragallinarum strains to be included in a vaccine, not only by their serotype. If field trials are successful, this methodology might and should be adopted in other regions to improve vaccine performance on a local/regional basis, especially where Infectious Coryza is prevalent.” (lines 442-447 of the revised manuscript).

Specific comments:

  1. Lines 29-34: Introduce Page and Kume serogroups here, it confuses readers who are not familiar with Avibacterium paragallinarum when these terms are used later in the text.

To improve understanding, the paragraph was edited to: “Page classified this bacterium into three serogroups A, B, and C, according to the plate agglutination test [2]. Later, these schemes were amplified to recognize nine Kume serovars: A‐1, A‐2, A‐3, A‐4, B‐1, C‐1, C‐2, C‐3, and C‐4 using hemagglutination inhibition tests [3,4]”. (lines 32-35 of the revised manuscript).

  1. Lines 160-162: Kindly add a description of the criteria used for the different scores

A new paragraph was added with the full description, as requested (lines 246-253 of the revised manuscript):

Evaluation of clinical signs

In all trials, clinical signs of each facial side were registered according to the following criteria: 0, no clinical signs; 1, light conjunctivitis; 2, facial inflammation, infraorbital sinusitis, and/or conjunctivitis and/or tears; 3, oedema and facial inflammation, infraorbital sinusitis, abundant nasal and eye secretion, and partially closed eye; 4, severe oedema and facial inflammation, abundant nasal and eye secretion, and totally closed eye. Scores of 0 and 1 were considered negative while scores of 2, 3, or 4 were considered positive.”  

  1. Lines 167-169; 183; 188, 214: The wording is confusing, how can clinical scoring be done on days 2, 5 and 7, and the birds also killed? If they were killed on day 2 they can’t be clinically scored on day 5 or 7…. I think the confusion lies in that the days of euthanasia have not been clarified. And then I think you mean that the bacteriology was performed directly on the day of euthanasia. Please add day of euthanasia (and humane-endpoint-criteria if used), and rewrite these sentences more clearly.

The text was edited as requested:

  1. Line 190: translate Spanish “y” to English “and”.

The text was edited as requested:

  1. Line 201: change “an” to “and”

The text was edited as requested:

  1. Line 202: remove “A”

The text was edited as requested:

  1. Lines 214-215: please change: “a diseased bird was considered if” to “a bird was considered diseased if”

The text was edited as requested:

  1. Line 245: correct “infection coryza”

The text was edited as requested:

  1. Table 2:  please add number of birds per group, at least in the table legend, and number of dead animals per group

The number of birds per group is figured in the description of the Table. Additional text “No deaths were registered“ was added at the end of the description” (now presented as Table S2).

  1. Table 6: please define “number of positive birds”.

The description of the table was modified and more information was added as requested:

“The birds were scored individually between 0 and 4 points for each facial side. A bird was considered positive if scored at least 2 points in one infra-orbital sinus.” (now presented as Table 2 of the revised manuscript).

Round 2

Reviewer 1 Report

Accepted

Author Response

Dear Reviewer. Thank you so much for your comments and your support in improving our manuscript.

Reviewer 2 Report

The authors have revised the manuscript correctly and have improved the content and presentation.

I have some concerns regarding the merge of Results and Discussion, but I can understand the viewpoint of the authors in presenting the results and commenting upon them at the same time. However, I recommend that a new subsection would be added at the end of this large merged section to recapitulate briefly the comments and to present a global overview of the findings in relation to potential clinical applications in the future.

After this change, the manuscript can be accepted.

Author Response

After the Results and Discussion section, there is another section Conclusions where a short summary of the methodology and the results are presented with the possible impact of the application of the findings. This paragraph was edited to better understand:

“The present work is the first that studied the pathogenicity of Av. paragallinarum strains isolated from poultry in Peru. A set of pathogenicity trials of these strains, regarding their capacity to cause clinical signs, horizontally infect susceptible birds, and their capacity to cause septicaemia and death was performed. A great variation among these parameters was observed, even within strains of the same serogroup. Afterwards, the most pathogenic strains for each serogroup were selected to formulate an experimental multivalent inactivated vaccine and a vaccination-challenge trial was performed. Homologous protection with satisfying levels of protection was observed, reducing clinical signs and significantly reducing the colonization of the infraorbital sinuses after infection, reducing the number of carrier birds.

The results of the present trials suggest an alternative methodology for the selection of Av. paragallinarum strains to be included in a vaccine according to their pathogenicity and not only by their serotype. Future trials should include a performance comparison of the experimental vaccine with commercial vaccines (positive control) and test the efficacy in a complete vaccination program in layers during production under controlled conditions or in field trials [69]. If field trials are successful, this methodology might and should be adopted in other regions to improve vaccine performance on a local/regional basis, especially where Infectious Coryza is prevalent. “

Reviewer 3 Report

Unfortunately, the response to the initial concerns related to the experimental design of this investigation were neither substantive nor adequate. Specifically, it was not clearly stated that each experimental exposure regardless of the experimental group and trial was performed in duplicate. Can that be clearly stated in the manuscript for each experimental group without qualification? Further, Figure 1S does not provide any substantive information, but is only a  superficial flow chart. In addition to the primary concern (above), rewrite the M&M section for clarity and simplicity.

Author Response

Dear Reviewer, we find it difficult to answer your request. In all trials, there were enough repetitions as each bird was considered as the experimental unit. The experimental treatments (vaccination, inoculation, and control) were applied to each bird individually and it is important to keep in mind that the birds from the experimental treatments were divided into two subgroups separated from each other only due to the capacity of each isolator and to make easier the experiment to follow up. As far as we understand, there was no need to separate them this way. Furthermore, despite this division into the two isolators, all birds were under the same experimental conditions during the whole time. Again, we find it difficult to understand your request for “duplicates” when there are much more repetitions for each treatment.

According to the comments that were received from you after the first submission, we have modified and adapted the manuscript to explain that clearly and we find no need to modify anything beyond. Furthermore, this matter was consulted before the beginning of the trials with our statistics advisor and again now.

Nevertheless, we would greatly appreciate it if you could suggest an alternative way to express it better in the manuscript and give us more details regarding what should be clarified in the M&M section. Please let us know how can we improve our manuscript.

Round 3

Reviewer 3 Report

During the development of the experimental design and methods, there are standards methods to determine, at least approximately, the number of "experimental units (EU)" (that is, individual animals) to be used in the course of the experiment(s) and the number of groups containing a certain percentage of the total EU. It is generally standard to use two groups (that is, duplicate groups), but often three or more groups, whereas each group contains the same number of EU. In that context, what were the methods used to determine the number of EU for each experiment/exposure and, to reiterate, the number of groups that contained an equal percentage of the total EUs? Please explain/provide the methodology information in detail.

Further, was the statistician that you mentioned in your response included as a co-author on the manuscript, since there was no Acknowledgment Section at the end of the manuscript.

Also, was the experimental design/methods reviewed by an intramural or extramural (consulting) Animal Care Committee such as an Institutional Animal Care and Use Committee (IACUC) or a similar Committee approved by the International Council for Laboratory Animal Science (ICLAS) prior to the initiation of the exposure experiments?

Author Response

Dear Reviewer

Trials 1, 2, and 3 were conducted based on previous works that described the pathogenicity of Avibacterium paragallinarum strains. In these papers [1–3], each experimental group consisted of 10 chickens. No explanation is given as to how they defined this number of birds. Nevertheless, in these works, the authors could not find statistical differences and therefore, we suggested that this number should be raised. According to the capacity of our facilities, we chose to duplicate it, allocating 20 chickens in two isolators with 10 in each of them, all under the same housing and rearing conditions.

In Trial 4, we estimated comparisons of means: the minimum infective dose that causes 100% of infection in the control group with 50% in treated groups. The calculation with 95% of confidence and a potency power of 90% gave a sample size of 11 chickens per group. This number is similar to other works that used 10 birds per group in challenge trials [4,5]. We decided to use 12 chickens per group and divided them into two isolators according to the age of the birds and the size of the isolators.

This information was added to the revised manuscript (lines 245-253).

In all our trials, the experimental unit was considered the chicken that was inoculated individually. The distribution of the chickens among the experimental groups was random (line 137 of the revised manuscript).

The first and last authors of this work are responsible for the experimental design and the definition of samples sizes. We also consulted with an external specialist from INTA (a member of the local IACUC) to make sure that our assumption that each chicken may be considered as the experimental unit and that the calculations were correct. We consider that there was no need for acknowledgments, as this was a general consulting that served for other experiments that were done before. 

All trials were presented and approved by the IACUC, as mentioned in the manuscript (line 266-270).

References

  1. Patil, V.; Mishra, D.; Mane, D. Virulence Pattern of Avibacterium paragallinarum Isolates Studied from Indian Field Condition. Int. J. Livest. Res. 2017, 1, doi:10.5455/ijlr.20170209071608.
  2. Chukiatsiri, K.; Sasipreeyajan, J.; Blackall, P.J.; Yuwatanichsampan, S.; Chansiripornchai, N. Serovar Identification, Antimicrobial Sensitivity, and Virulence of Avibacterium paragallinarum Isolated from Chickens in Thailand. Avian Dis. 2012, 56, 359–364, doi:10.1637/9881-080811-Reg.1.
  3. Byarugaba, D.K.; Minga, U.M.; Gwakisa, P.S.; Katunguka, E.R.; Bisgaard, M.; Olsen, J.E. Virulence Characterization of Avibacterium paragallinarum Isolates from Uganda. Avian Pathol. 2007, 36, 35–42, doi:10.1080/03079450601102947.
  4. Jacobs, A.A.; van den Berg, K.; Malo, A. Efficacy of a New Tetravalent Coryza Vaccine against Emerging Variant Type B Strains. Avian Pathol. 2003, 32, 265–269, doi:10.1080/0307945031000097859.
  5. Soriano-Vargas, E.; Longinos, G.; Fernandez, R.; Velásquez, Q.; Ciprián, C.; Salazar-García, F.; Blackall, P.; VE Soriano, G.L.R.F. Virulence of the Nine Serovar Reference Strains of Haemophilus paragallinarum. Avian Dis. 2004, 48, 886–889, doi:10.1637/7188-033104r1.
